# Process and Eco-Environment Impact of Land Use Function Transition under the Perspective of “Production-Living-Ecological” Spaces—Case of Haikou City, China

**DOI:** 10.3390/ijerph192416902

**Published:** 2022-12-16

**Authors:** Wenxing Du, Yuxia Wang, Dingyi Qian, Xiao Lyu

**Affiliations:** 1College of Geography and Environmental Science, Hainan Normal University, Haikou 571158, China; 2Key Laboratory of Earth Surface Processes and Environmental Change of Tropical Islands, Hainan Normal University, Haikou 571158, China; 3School of Humanities and Law, Northeastern University, Shenyang 110169, China; 4Key Laboratory of Coastal Zone Development and Protection, The Ministry of Land and Resources, Nanjing 210024, China

**Keywords:** production-living-ecological spaces, land use function transition, land use transfer matrix, eco-environment quality index, ecological effect

## Abstract

Land use function transition can change the eco-environment. To achieve an “Intensive and efficient production space, moderately livable living space, and beautiful ecological space”, the ecological effects of land use function transition in the context of ecologically fragile areas and rapidly developing areas of socio-economic importance need to be studied. In this study, from the perspective of “production-living-ecological” spaces, we calculated the index of regional eco-environment quality, positive and negative effects of eco-environment impact, and the ecological contribution rate and analyzed the driving factors. We found the following: (1) The production space was greatly compressed, living space was expanded, and ecological space was significantly squeezed. Haikou underwent a rapid transformation from an agriculture-dependent city to an industrial city. Land supply for urban and rural living was guaranteed by the Chinese land management department. However, Haikou prioritized economic development over environmental protection. (2) The regional eco-environment quality index decreased from 2009 to 2018. The expansion of pasture-based ecological spaces is important for improving the quality of the eco-environment, and the reduction of forest ecological space strongly influences the deterioration of the eco-environment. (3) Resource base, historical level of utilization, suitability of land, the ecological value potentiality, and regional policies greatly affected land use function transition and its eco-environment. (4) Refining the *planning of territorial space*, comprehensively improving land and resources, and reforming the *rural land system* greatly influenced policy guidance and technical regulation for coordinating “production-living-ecological” spaces and improving the regional eco-environment. In this study, we tested the effect of regional policy regulation on land use function transition and provided a reference for coordinating “production-living-ecological” spaces.

## 1. Introduction

Land use function (LUF) is widely investigated around the world [1,2,3,4]. Land use function transition (LUFT) has also become a popular research topic [5,6]. The concept of land use transition was proposed by the British geographer Grainger, based on the forest transition hypothesis [7]. In the early 21st century, Hualou Long introduced the concept into China to provide a new perspective on and approach to the study of land use/land cover change (LUCC), which refers to the change in land use morphology over time corresponding to the transformation of economic and social development [8]. In the last 20 years, based on several studies, researchers have divided land use morphology into two types: dominant morphology and recessive morphology [9,10]. LUFT is a part of land use recessive morphology transition [11]. Several researchers have investigated LUFT and its effects on cultivated land [12,13,14], rural residential land [15,16], and urban-rural construction land [17]. Concerning pattern analysis, several researchers have described the spatiotemporal evolution of LUFT [18,19] and clarified the pattern formation rules [20,21]. Concerning studies on the dynamic mechanism, some researchers have qualitatively and quantitatively analyzed the various driving factors of LUFT, including natural resource endowment, socioeconomic conditions [22], stakeholder behavior [23], land engineering and technology of comprehensive improvement [24,25], land system and policies [26,27,28,29], etc. Regarding studies on the effects on and responses of LUFT, the impact on the eco-environment is the most significant [30,31,32,33,34,35,36]. Several studies have investigated the eco-environmental effects of “production-living-ecological” functions transition (PLEFT) [37,38,39,40,41,42,43,44]. Current studies on LUFT cover a wide field. However, studies on the pattern evolution rules of production-living-ecological spaces (PLES), including those on all types of lands, covering whole urban and rural areas, need to be conducted to identify the positive and negative effects of PLEFT on the eco-environment.

The goal of the construction and development of ecological civilization was described in the 18th National Congress of the Communist Party of China (2016). The goal of “intensive and efficient production space, moderately livable living space, and beautiful ecological space” indicates the coordination of PLES. The goal reflects the land use mode transformation from supplying land for meeting the demands of rapid social and economic development to the coordination of production, life, and ecology in China. Haikou city of Hainan province started building an international tourist island in 2010. The “Haikou-Chengmai-Wenchang” integrated development economic circle was established in 2015, and a free trade pilot zone was established in 2018. To adjust to the national macro-policies, the Hainan province of China quickly entered a new period of social and economic development. Specifically, since Haikou is the political, economic, and cultural center of Hainan province, it strongly influences the construction of Hainan’s International Tourist Island and Free Trade Port. Haikou has accelerated urbanization and industrialization and focused on the construction of ecological civilization. The influx of people, the accumulation of capital and industry, and the expansion of construction land are big challenges to the protection and improvement of the eco-environment. Studies focusing on the key area of national strategic adjustment and ecologically valuable areas, such as Haikou city of China, are few. It is necessary to conduct studies to test the effects of the implemented policies, based on which the driving factors of such policies might be recommended.

To summarize, in this study, we investigated the effect of LUFT in Haikou on the eco-environment. The paper is based on the mutual conversion of “production-living-ecological” spaces (PLES). Our aims were as follows: (1) To depict the scale and direction of the conversion of PLES by analyzing the pattern evolution rules and driving factors; (2) To determine the regional eco-environment quality index for distinguishing the positive and negative effects of LUFT on the eco-environment; (3) To suggest targeted policies for improving the quality of the eco-environment.

## 2. Study Area and Data

### 2.1. Study Area

Haikou is a southern coastal city located in the tropics and is also known as the “Coconut City”. It is a prefecture-level city and the provincial capital of Hainan Province. It is a national strategic hub of the “belt and road”, and an important node of Beibu Gulf metropolitan. It is located between 19°31′~20°04′ N and 110°07′~110°42′ E in the north of Hainan Island, adjacent to Wenchang in the east, Chengmai in the west, Dingan in the south, and Qiongzhou Strait in the north. It is the political, economic, technological, and cultural center of Hainan Province (Figure 1). This area has a surplus of natural resources, including a land area of 2289.51 km^2^, a sea area of 861.44 km^2^, and a coastline of 136.23 km. At the end of 2018, the urbanization rate was 78.6%, and the gross regional product (GDP) was 151.05 billion yuan, which was the highest among all counties and cities in Hainan Province. The total area of production land (PL) was 1383.52 km^2^, the total area of living land (LL) was 354.52 km^2^, and the total area of ecological land (EL) was 551.48 km^2^, accounting for 60.43%, 15.48%, and 24.09%, respectively, of the total land. Xiuying, Longhua, Qiongshan, and Meilan are under the jurisdiction of the city, covering an area of 495.10 km^2^, 303.20 km^2^, 928.62 km^2^, and 562.59 km^2^, accounting for 21.62%, 13.24%, 40.56%, and 24.57%, respectively, of the total area of the city.

### 2.2. Data Sources and Processing

The Second National Land Survey in China was conducted in 2008. From 2009 to 2018, we obtained the land use data of Haikou from the annual change database of land use provided by the Hainan Provincial Management Department of Natural Resources. According to the actual functions to meet the various needs of humans, land use functions can be classified into three first-level land use types and eight second-level land use types. The three first-level land-use dominant functions included production land, living land, and ecological land. The eight second-level land use functions were divided into agricultural production land, industry-mining production land, urban living land, rural living land, forest ecological land, grassland ecological land, aquatic ecological land, and other types of ecological land. Based on the findings of published studies [37,38,45,46] and the requirement of Chinese Territorial Spatial Planning, we proposed the classification of dominant land use functions and their corresponding eco-environment quality index for the land use characteristics of Haikou (Table 1).

Notes in Table 1 are as follows: (1) The role of the reservoir is to solve the seasonal and structural imbalance of surface water in specific areas. The biggest function of the reservoir is to provide irrigation water for agricultural production. Ordinarily, agricultural water consumption is far greater than living water consumption and industrial production water consumption. So, in this paper, the reservoir water surface in the Chinese Second National Land Survey database was merged into agricultural production land. This classification was also an important adjustment in the integration of the Chinese Third National Land Survey into the land category of Chinese Territorial Spatial Planning. (2) Although the garden land had an ecological function, the relative area of the garden was not large, and it was mainly to obtain agricultural income for economic crops in Hainan Province, so it was included in agricultural production land rather than grassland ecological land.

## 3. Research Methods

### 3.1. Research Framework

Land use function transition (LUFT) is affected by several factors, such as natural resource endowment, national macro-strategy adjustment, socio-economic conditions, regional systems and policies, science and technology, etc. The ecological effect is an important parameter as it reflects the influence of regional macro-micro policies on land use. Many factors affect the types and ways of land use, causing the transfer of PLES and the transition of LUF, which in turn can lead to changes in the ecological effect [47,48].

Population, enterprises, capital, and technologies tend to move to and accumulate in Haikou city of Hainan Province as national socio-economic development policies and strategies are adjusted, such as the establishment of a Free Trade Port and rural land system reform in Hainan Province, resulting in the conversion of different land use types. Production land occupied by living land, ecological land occupied by living land, and production land et cetera are examples. Therefore, living space oriented by living land would compress production space oriented by production land. Also, there would be a transfer from production space to living space or others. In addition, with the spatiotemporal transfer and optimization of production space, living space, and ecological space (PLES) based on resource suitability evaluation and historical consumption of land resources, a series of responses to these policies and strategies will emerge, such as transition in the land use function (LUFT), which is also determined by land use type, and changes in eco-environment quality. All of them can be attributed to various factors, such as internal conversion between agricultural and industrial-mining production land, conversion of production land to living land, and so on. Regional eco-environment quality indices were used to compare the positive and negative effects of LUFT on the environment that corresponded with socioeconomic development in specific regions over time. The LUFT is primarily determined by changes in the structure of land use types over time, including the quantity and spatial distribution of attributes. The transfer of PLES denotes changes in attributes such as regional eco-environment quality and the positive or negative effects of LUFT on the environment.

In this study, data were obtained from the annual land use change database. Based on the land use characteristics of Hainan, we divided the production-living-ecological land use functions, constructed the land use transfer matrix of PLES, described the conversion characteristics of PLES, analyzed the positive and negative effects of LUFT on the regional eco-environment, identified the ecological risks, and determined the factors that drive policies. We tested the eco-environmental effects of the regionally differentiated land control policies and proposed countermeasures and suggestions for coordinating PLES (Figure 2). The findings of this study might provide valuable information and contribute to the construction of an ecological civilization in Hainan.

### 3.2. Land Use Transfer Matrix of LUFT

The land use transfer matrix was constructed to arrange the transfer area of each land type in the form of a matrix. The matrix can help to determine the direction and scale of one land type transferred to other types. In the transfer matrix, the characteristics of land use change can be intuitively expressed. It is a fundamental method for analyzing land use change. The dominant functional land types of land use were divided, based on local conditions. The land types in the land use annual change database were merged into the first-level and second-level land use function categories. Finally, the land use transfer matrix of LUFT was constructed. In the matrix, we analyzed the transition process of land use function in terms of scale, structure, space, and land type. It was calculated as follows:(1)$$S_{ij}=\left[\begin{array}{cccc} S_{11} & S_{12} & \cdots & S_{1n}\\ S_{21} & S_{22} & \cdots & S_{2n}\\ \cdots & \cdots & \cdots & \cdots \\ S_{n1} & S_{n2} & \cdots & S_{nn} \end{array}\right]$$

Here, *S* represents the area, *i* and *j* represent land use function types at the beginning and end of the research period, respectively, and *n* represents the number of types of land use functions.

### 3.3. Regional Eco-Environment Quality Index of LUFT

We found that different land types determined different land-use functions and led to differences in the quality of the regional eco-environment. By determining the regional eco-environment quality index, the overall changes in the eco-environment quality caused by LUFT in a certain area in different periods can be quantified. The regional eco-environment quality index refers to the eco-environmental quality and its area ratio of different land types in the classification system of PLES [37,38,39,40,41,42,43,44]. It was calculated as follows:(2)$$EV_{t}=\overset{n}{\underset{i=1}{\sum }}\frac{S_{ki}}{S_{k}}R_{i}$$

Here, *EV_t_* represents the regional eco-environment quality index in the *t* period; *S_ki_* represents the area of land-use type *i* in the *t* period; *S_k_* represents the total area of this region; *R_i_* represents the eco-environment quality index of the land-use type *i*; *n* represents the number of land-use types in the region.

### 3.4. Ecological Contribution Rate of LUFT

Ecological contribution rate refers to the changes in the regional eco-environment quality caused by changes in the dominant function of land use and can be used to quantify the impact on the regional eco-environment caused by the mutual conversion of various functional lands. It can be analyzed from the perspective of positive and negative effects (Table 2). Hence, it is suitable for determining the leading factors that cause changes in the regional eco-environment, separating the main type of LUFT that affects changes in the quality of the eco-environment. The ecological contribution rate can be calculated as follows:(3)$$LEI=100\left(LE_{1}-LE_{0}\right)LA/TA$$

Here, *LEI* represents the ecological contribution rate of LUFT; *LE*_0_ represents the initial value and *LE*_1_ represents the final value of the eco-environment quality indices reflected by certain LUFTs; *LA* represents the area of this transition type; *TA* represents the total area of this region.

## 4. Results and Analysis

### 4.1. The Scale Change of PLES in Haikou

(1) The scale of production space decreased sharply, and a transition from agricultural dependence to industrial development was recorded.

From 2009 to 2018, the production space in Haikou decreased, with a total reduction of 2031.54 ha, indicating a decrease of 1.45%. Within the production space, the scale of industry-mining production space increased by 1942.71 ha, indicating an increase of 19.47% (Table 3), and the agricultural production space decreased by 3974.25 ha, indicating a decrease of 3.05%. The scale of production space in Xiuying, Longhua, Qiongshan, and Meilan, which are under the jurisdiction of Haikou, decreased by 1.4%, 1.6%, 1.53%, and 1.23%, respectively. Within the production space, the scale of agricultural production space decreased and that of industrial production space increased substantially in these four districts. The homogeneity of total scale change in production space and change in its internal structure showed that Haikou city and its four districts underwent a rapid transition from agricultural dependence to industrial development during the research period.

(2) The Scale of living space, including urban and rural living spaces, increased.

From 2009 to 2018, the scale of living space in Haikou increased by 4142.8 ha, indicating a total increase of 13.23% and an average annual increase rate of 1.47%. The scale of urban living land increased by 3124.38 ha, indicating an increase of 14.92%, while the scale of rural living space increased by 1118.42 ha, indicating an increase of 10.13%. The living spaces of Xiuying, Longhua, Qiongshan, and Meilan increased by 19.24%, 13.19%, 9.25%, and 11.59%, respectively, and the urban and rural living spaces increased. The expansion of the total area of living space and the expansion of both urban and rural living spaces showed that rapid development and urbanization occurred in Haikou, which suggested that the development of urban and rural land markets was coordinated. These findings also showed that the Rural Planning of Hainan Province protected the qualification rights and property rights of farmers.

(3) The scale of ecological space decreased significantly, and the protection of ecological land weakened.

From 2009 to 2018, the scale of ecological space decreased by 1587.1 ha, indicating a decrease of 2.80%. The reduction of ecological space was greater than the reduction of production space. Within ES, the scale of forest ecological land, aquatic ecological land, and other ecological land decreased by 3.76%, 1.52%, and 12.72%, respectively. The abnormal land type was grassland ecological land, which covered an area of 3.51 ha in 2009 but increased by 901.8 ha at the end of 2018. The ecological space scale of Xiuying, Longhua, and Meilan decreased by 1.67%, 4.15%, and 3.08%, respectively, while that of Qiongshan increased by 2.09%. Within ES, the scale of forest ecological land, aquatic ecological land, and other ecological land decreased, but the pasture ecological land increased considerably. The change in the homogeneity of ES in Haikou and its districts showed that the protection of ecological land in Haikou and its three districts were relatively weak, probably due to rapid urbanization and modern industrialization, and they prioritized development over environmental protection. The ecological value increased only in Qiongshan. Although the overall economic development of Qiongshan was slower than that of the other districts, its ecological value and significance were the highest among the four regions in Haikou.

(4) The tension in living space was high, and the resilience of production space and ecological space was inadequate.

From 2009 to 2018, the living space (LS) of Haikou increased by more than 13% in 10 years, with an average annual growth rate of 1.25%. Simultaneously, production space (PS) and ecological space (ES) decreased by 1.45% and 2.8%, respectively (Figure 3). In Xiuying, Longhua, Qiongshan, and Meilan, urban and rural living spaces increased. Although LS increased, PS and ES decreased in these four districts (Figure 4). Due to the expansion of LS, both PS and ES were compressed by LS. The ratio of shrinking PS to increasing LS was 1.16. The ratio of shrinking ES to increasing LS was 2.24. The change in the homogeneity of the PLES in Haikou and its jurisdiction showed that the resilience of the PS and ES of Haikou was inadequate, and thus, LS increased the pressure on PS and ES. Especially, the gap between the further development of the ES and the goal of ecological civilization construction was high. Our results also showed that under the dual policy guidance of International Tourism Island Construction and Free Trade Port Construction, the socioeconomic development of Hainan includes population concentration and real estate industry orientation.

### 4.2. Characteristic Analysis of the Conversion of PLES

From 2009 to 2018, the spatiotemporal pattern evolution of PLES in Xiuying, Longhua, Qiongshan, and Meilan of Haikou showed the following characteristics.

(1) The production space (PS) in all four districts was occupied by LS, and this trend was the most prevalent in Xiuying. The PS decreased from the east and the west to the central-western region. Regional policies have played a significant role in guiding LUFT.

During the research period, 2333.32 ha of PS was occupied by LS in Haikou (Table 4), including 819.8 ha, 404.49 ha, 499.1 ha, and 609.9 ha in Xiuying, Longhua, Qiongshan, and Meilan. In these four districts of Haikou, 35.13%, 17.34%, 21.39%, and 26.14% of the PS was converted into LS, and the contribution levels were level I, level IV, level III, and level II, respectively (Figure 5). The decrease in PS and the increase in LS in Xiuying were the highest, followed by similar changes in Qiongshan, Meilan, and Longhua. The PS decreased considerably at both ends and slightly in the middle, contracting from the east and the west to the middle-western region.

Xiuying lies in the west of Haikou and has abundant coastal resources. As it is the sub-central urban region of Haikou, the development of this region focused on urbanization and coastal tourism. Owing to huge financial investments and rapid construction, government departments have been relocated and concentrated. Urban public service infrastructure is well-established, and the real estate industry flourishes in this region. Thus, Xiuying is a key development district in Haikou, as it promotes the expansion of LS. The requirements for the coordinated development of the east and west coasts were also proposed. The development of the Jiangdong Free Trade Zone also promoted the rapid expansion of LS in Meilan. The conversion of PS to LS in Xiuying, Longhua, Qiongshan, and Meilan reflects the implementation of regional policies, and the adjustment of relevant planning strongly influences space conversion and expansion. The conversion of PS to LS also reflects that Xiuying and Meilan, under the dual policy of International Tourism Island construction and Free Trade Port construction, have undergone significant changes in population concentration and real estate industry orientation.

(2) The reduction in the agricultural production space in Xiuying, Longhua, Qiongshan, and Meilan was accompanied by the expansion of industrial-mining production space, among which, Xiuying and Meilan showed significant changes. The industrial-mining production space increased from the center to the east and west. The natural resource basis and the macro-policy adjustment jointly laid the foundation for LUFT.

During the research period, 1168.28 ha of agricultural production space in Haikou was occupied by industrial-mining production land, which was 518.06 ha, 101.49 ha, 116.66 ha, and 432.07 ha of area in Xiuying, Longhua, Qiongshan, and Meilan, respectively. The contribution rate of the four districts was 44.34%, 8.69%, 9.99%, and 36.98 (levels I, IV, III, and II, respectively). The sub-region contribution level of industrial-mining production space conversion was similar to that of PS conversion (Figure 5). The expansion of the industrial-mining production space with the reduction of agricultural production space in Xiuying was the largest, followed by Meilan, Longhua, and Qiongshan, which had a similar level of space conversion. The tension of PS was high at both ends of Haikou and low in the middle. Industrial-mining production space increased from the middle to the east and west.

After many years of construction, new industries were established in Xiuying. Transportation and connectivity via railways, highways, airports, ports, and terminals are well-developed, and the area is highly industrialized. Meilan has also shown rapid progress, with industrial development and the construction of the transportation network. Longhua has a smaller space for development and a lesser potential. Qiongshan is an old district and has slow development. Xiuying, Longhua, Qiongshan, and Meilan all show the same pattern of a decrease in agricultural production space and an increase in industrial-mining production space. This reflects the simultaneous transition from dependence on agriculture to the development of industries and mining activities in these four districts. This transition also reflects that the base number of natural resources, historical consumption of resources, and the adjustment of regional development orientation and policies have jointly laid the foundation for LUFT.

(3) An increase in rural living space is accompanied by an increase in urban living space, but the tension of urban living space is higher than that of rural living space in Xiuying, Longhua, Qiongshan, and Meilan, among which, Meilan has the highest tension. The expansion of urban living space has led to a decrease in rural living space, expanding from the east to the central and western regions. The adjustment of regional macro-policies is an important driving force for LUFT.

During the research period, urban living land increased by 14.92%, and rural living land increased by 10.13% in Haikou. Approximately 36.76 ha of rural living land was occupied by urban living land. Xiuying, Longhua, Qiongshan, and Meilan occupied 2.05 ha, 7.85 ha, 8.65 ha, and 18.21 ha, respectively. The contribution rate of the rural living space encroached by urban living space in the four districts was 5.58%, 21.35%, 23.53%, and 49.54%. Their contribution level was III, II, II, and I, respectively (Figure 6). Meilan had the greatest expansion of urban living space and reduction of rural living space, followed by Longhua, Qiongshan, and Xiuying in descending order. The urban living space squeezed the rural living space and expanded from the east to the central and western regions.

Meilan is the seat of the Hainan Provincial Party Committee and the provincial government. It is the political, economic, and cultural center of Hainan Province and the main urban area of Haikou. It is located in the east wing of Haikou. Meilan is also the eastern core of the “Haikou-Chengmai-Wenchang” integrated development economic circle. Due to the presence of three large industries, including tourism, the modern service industry, and the high-tech industry, this region has become popular, leading to an increase in population and business. This has increased the demand for living land and has accelerated the expansion of living land. However, the speed of urbanization is considerably higher than that of urban-rural integration, and the tension of urban living space is higher than that of rural living space. Urban living space occupies rural living space in Xiuying, Longhua, Qiongshan, and Meilan. This not only reflects the vitality of urban development and the sluggishness of rural development but also that the degree of urban-rural integration in Meilan is not high, while the urban-rural land allocation in Xiuying is more coordinated.

(4) Among all types of ecological spaces, only the grassland ecological space increased in Xiuying, Longhua, Qiongshan, and Meilan. The increase was mainly due to the occupation of agricultural production space and other ecological spaces. Agricultural production space was occupied in Qiongshan and Meilan, while other ecological spaces were occupied in Xiuying and Longhua. The suitability of land resources is the most basic condition to determine LUF.

The grassland ecological land space increased 257 times in 10 years, while the forest ecological space, aquatic ecological space, and other ecological space decreased by 3.76%, 1.52%, and 12.72%. Two main sources were converted into the grassland ecological space: the agricultural production space and other ecological spaces. The agricultural production space contributed 792.52 ha, accounting for 87% of the ecological space expansion source in Haikou, while other ecological spaces contributed 115.27 ha, accounting for 12.6% of the expansion source in Haikou. In Xiuying and Longhua, the expansion of grassland ecological space occurred due to the conversion of other ecological spaces, accounting for 76.5% and 96.6%, respectively. In Qiongshan and Meilan, almost 100% of the expansion of grassland ecological space was due to the conversion of agricultural production space.

In Qiongshan, local villagers conceived a new eco-friendly way to get rid of poverty, which involved the development of an artificial grassland planting industry. In 2014, they built a grassland demonstration base according to local soil quality. Combining the regional poverty alleviation policies with the responsiveness of village organizations, Qiongshan, an old district of Haikou, though slower in socio-economic development than the other three districts, lowered its ecological risk considerably. Natural resource endowment (especially, soil quality) determines the possibility of LUFT. The improvement of the national targeted poverty alleviation policy also promoted the emergence of the land-use ecological function.

### 4.3. Characteristics of Changes in Eco-Environment Quality

During the study period, ES in Haikou decreased by 2.80%, and the eco-environmental quality index decreased from 38.15 to 37.71, indicating a decrease of 1.15% (Figure 7). The area of ES occupied by PS and LS was 2283.56 ha. The agricultural production function of land use decreased, and the industrial production function increased. The urban and rural living functions increased. The conversion of PLES decreased the ecological functions of land use, which might be associated with ecological risks.

In the study period, among the four districts, the eco-environment quality index increased by 0.09% only in Qiongshan. The index in the other three districts decreased. The indices decreased by 2.89% in Xiuying, 1.55% in Meilan, and 1.06% in Longhua. Due to rapid urbanization and industrialization, Xiuying had greater ecological risks. Qiongshan, the traditional old district, had different ecological values. Regarding the improvements in the eco-environment quality of Haikou, Qiongshan had a contribution of level I, and Longhua, Meilan, and Xiuying had a contribution of levels II, III, and IV, respectively (Figure 8). These results further confirmed the above analysis of the contribution of the sub-region to the conversion of ES in Haikou.

The forest ecological space in Qiongshan was large and less occupied by other land types during the study period. However, the grassland ecological space increased slightly from 2014 to 2018. These two types of ecological space strongly affected the regional eco-environment quality. Therefore, for all years, the eco-environment quality index of Qiongshan was significantly higher than that of the other three districts. The forest and grassland ecological spaces in Xiuying and Meilan were not large. The forest ecological space occupied the largest area in Xiuying, while the grassland ecological space did not increase considerably among the four districts. Thus, the greatest decline in environmental quality occurred in Xiuying (Figure 9). The area of the PLES in Xiuying, Longhua, Qiongshan, and Meilan indicated the eco-environment quality of Haikou, and the conversion of PLES directly changed the eco-environment quality of Haikou.

### 4.4. Positive and Negative Effects of the Eco-environment Impact

In the study period, 7593.26 ha of land was converted, accounting for 3.32% of the total area of Haikou. The area of conversion in Xiuying, Longhua, Qiongshan, and Meilan accounted for 33.9%, 15.1%, 21.5%, and 25.5%, respectively. The negative effect of the eco-environment impact caused by LUFT was 0.64, and the positive effect was 0.26 (Table 5). Only in Qiongshan the positive effect of the eco-environment impact led by LUFT was greater than the negative effect. The negative effect was considerably greater than the positive effect in Xiuying, Longhua, and Meilan (Figure 10). Qiongshan contributed to improving the overall eco-environment quality of Haikou, while the other three districts had a negative contribution. For rapid development, Xiuying and Meilan focused more on industrial land supply than on the protection of land use ecological function. Land use in Qiongshan was not high, but protection of the land use ecological function was extremely effective. Longhua occupied the smallest area, the conversion area of PLES in Longhua was the smallest, and the positive and negative effects of LUFT on the eco-environment quality were also the smallest. Thus, it contributed the least to the overall improvement of the eco-environment quality in Haikou. The level of resources and their internal structure were used to determine the background of the regional eco-environment quality, and the direction of PLES conversion determined its contribution rate to the improvement of the regional eco-environment quality.

The grassland ecological space in Haikou occupied 907.79 ha of other ecological spaces and agricultural production spaces, and its expansion nearly doubled. The contribution of the expansion to the positive effect of the eco-environment change was as high as 79.4% in Haikou, and 60%, 97%, and 76% in Longhua, Qiongshan, and Meilan, respectively. That expansion of the grassland ecological space considerably improved the eco-environment quality of Haikou. It also considerably improved the eco-environment in Longhua, Qiongshan, and Meilan, especially in Qiongshan (Table 6). The encroachment of industrial-mining production space by agricultural production space positively affected the eco-environment quality. The contribution rate to the improvement of the eco-environment quality in Xiuying reached 24%, which was higher than the contribution rate for the same index of the other three districts. This showed a positive signal; the reclamation of construction land in Xiuying was considerably higher than that of the other three districts. Xiuying found ways to redevelop and utilize inefficient and idle construction land, and thus, considerably improved the eco-environment of the region.

The forest ecological space was found to be occupied by 1531.64 ha of production space and living space in Haikou; thus, it was reduced by 98.6%, contributing 72% to the negative impact on the eco-environment quality. Forest ecological space was occupied, resulting in negative effects on the eco-environment quality in Xiuying, Longhua, Qiongshan, and Meilan. The contribution rate to the deterioration of the eco-environment quality was greater than 67%. Thus, the reduction of forest ecological space strongly influenced the decline in the eco-environment quality in Haikou and significantly contributed to the deterioration of the eco-environment in Haikou. The agricultural production space was also occupied, which had negative effects on the eco-environment. It contributed around 23% in Xiuying, Longhua, Qiongshan, and Meilan. The reduction of agricultural ecological space was also responsible for the decrease in the eco-environment quality in Haikou.

## 5. Discussion and Conclusions

### 5.1. Discussion

We found that Haikou underwent a rapid transformation from an agriculture-dependent city to an industrial-development city, and it prioritized economic development over environmental protection. The expansion of grassland ecological space strongly influenced the quality of the regional ecological environment, and the decrease in forest ecological space contributed to the deterioration of the ecological environment. The standardization and refinement of Territorial Space Planning, comprehensive improvement of land, and reformation of the rural land system played a major role in policy guidance and technical regulation for coordinating PLES and improving the regional ecological environment.

Regarding the driving mechanism and the effect of LUFT, several researchers have found that returning farmland to forest and grassland might not significantly alter LUF [51]. However, we found that the expansion of grassland ecological space improved the quality of the regional ecological environment, and the decrease in forest ecological space led to the deterioration of the ecological environment. In China, the conversion of farmland to forest land and grassland improved the regional ecological environment. According to some researchers, policies act on the types and ways of land use, which leads to LUFT and causes the chain-change of land use effect [52,53]. In this study, we comprehensively analyzed the influencing factors of regional LUFT from the perspective of natural resource endowment, national macro-development strategy, social and economic development conditions, regional institutional policies, scientific and technological level, etc., and the corresponding policy suggestions were specifically for the region. By simulating the scenario, a study showed that the reduction of forest area in Hainan Island from 2010 to 2040 might be the main reason for the deterioration of the ecological environment, and the deterioration of the ecological environment of the whole island was found to be higher than improvement of the environment [54]. However, we found that although the ecological effect of land use in Haikou decreased from 2009 to 2018, in some areas, such as the Qiongshan district, it increased slightly, and the ecological effect of LUFT showed regional differences and staged differences.

Our study was characterized by the division of production-living-ecological land and the conversion of PLES based on the local characteristics of Hainan. We associated LUFT with many driving factors and their positive and negative effects on the ecological environment. The findings of our study might help to test the effect of regional policies on land use and act as a reference for the construction of the Hainan Free Trade Port guided by ecological civilization.

We found that LUFT has a comprehensive impact on regional society, economy, ecology, and civilization. We only tested the effects of macro-micro policies on socio-economic development from the perspective of ecological effects of LUFT in specific regions, and our study had some limitations. Future studies should focus on the comprehensive effect and superposition effect of LUFT.

### 5.2. Conclusions

By calculating the land use transfer matrix of PLES, we analyzed the regional eco-environmental quality index, positive and negative effects of eco-environmental impact, ecological contribution rate, and LUFT and its eco-environmental impact in Haikou from 2009 to 2018. Based on our findings, we concluded the following:(1)Living space (LS) occupied the most area in PLES of Haikou, accounting for more than 60% of the total area. Among the four districts studied, Qiongshan accounted for 30%, while Longhua had the lowest, accounting for 6%. In the study period, the scale of PS decreased significantly, shrinking from the east and west to the central-western region. Within PS, the industrial-mining production space increased significantly, while the agricultural production space decreased significantly. A rapid transformation from dependence on agriculture to industrial development occurred in Haikou. The reduction of agricultural production space was accompanied by the expansion of industrial-mining production space, especially in Xiuying and Meilan. The industrial-mining production space expanded from the middle to the east end and west. The natural resource base superimposed on the macro-policy adjustment laid the foundation for LUFT.(2)Production space (PS) occupied the least area in Haikou, accounting for 15% of the total area. In the study period, the scale of LS increased considerably. The expansion of rural living space was accompanied by the expansion of urban living space, but the increase in urban living space was higher than that of rural living space, especially in Meilan. Urban living space encroached on rural living space and expanded from the east to the mid-west. During rapid social and economic development, the historical increase in LS in Meilan was small, and the potential to increase was high. We did not find an imbalance between urban and rural living land supply in the four districts, and the allocation of land resources between urban and rural areas was coordinated. The rural land system reform effectively protected the property rights and interests of the farmers. The historical foundation of resource use can partly determine the prospect and ability of LUFT.(3)The scale of ES in Haikou accounted for 24% of the total area. The decrease in ES was significant in the study period. The ES decreased in Xiuying, Longhua, and Meilan but increased in Qiongshan. The grassland ecological space was the only type of ecological space that increased in size. Its expansion was mainly due to the conversion of agricultural production space and other ecological spaces. We found that with rapid urbanization and industrialization, the protection of ecological land had weakened; Haikou prioritized development over environmental protection. Although the increase in urban living space and industrial-mining production space in the old district of Qiongshan was small and slow, the region maintained and created its ecological value. The suitability of resources is a natural basis for LUFT. Determining the functions of the planned land space is important for maintaining and improving the regional land use ecological function.(4)In the study period, the ES of Haikou was occupied by PS and LS, and the eco-environment quality index decreased. However, the positive effect of LUFT on eco-environment quality was greater than the negative effect in Qiongshan, which enhanced the eco-environment quality of Haikou. For rapid development, Xiuying and Meilan prioritized land supply for the industry, ignoring the protection of ecological function. Longhua had the smallest land area, the smallest conversion of PLES, and the smallest positive and negative effects on the eco-environment of LUFT. The level of resources not only determined the regional eco-environment quality but also regulated the improvement of the eco-environment quality in Haikou.(5)The increase in grassland ecological space improved the eco-environment quality in Longhua, Qiongshan, and Meilan. The reduction of forest ecological land and the decrease in forest ecological space decreased the eco-environment quality, which in turn caused the deterioration of the eco-environment. The decrease in grassland ecological space was the initial cause of the deterioration of the eco-environment in Haikou. The decrease in agricultural production space was the second cause. The utility and protection of forest land and grassland can significantly contribute to the maintenance and improvement of land use ecological function and effectively enhance the regional eco-environment quality.

## 6. Policy Implications

To improve eco-environment quality in Haikou, the following policies are recommended:

(1) For constructing the Free Trade Port, although, various policies have been implemented for promoting socio-economic development, the maintenance of land ecological function should not be ignored. Specifically, forest ecological land should be strongly protected by prohibiting indiscriminate logging and deforestation and preventing natural disasters, such as forest fires. Economy-intensive land use by enterprises in industrial parks should be started, and the land use scale of enterprises should be strictly investigated and evaluated. “Enclosure-style” extensive land use and occupation of forest land, grassland, and agricultural production land for industrial-mining land should be prevented. Strict rules should be implemented for the protection of arable land and ecological land and to only construct houses for living. The development of the real estate industry should be strictly regulated and prevented from occupying agricultural production space and ecological space.

(2) Before conducting Territorial Space Planning at all levels, “land suitability evaluation and land resource carrying capacity evaluation” should be performed. The red line of ecological protection should be drawn by performing field surveys and investigations, and the scale and scope of ecological land should be determined. Based on the main function division system of land use, the principle of hierarchical control of planning indicators should be followed for preparing the Territorial Space Planning at all levels. Specifically, for village planning, a standard area of 175 m^2^ of rural homestead in Hainan Province should be strictly implemented. The indicators for constructing land of different village types should be determined based on local conditions. These rules should be strictly implemented to prevent the disorganized expansion of rural construction land and the formation of new points to start construction.

(3) For promoting the Rural Land System Reform in China, including land requisition, rural collective construction land entering the market, and rural residential land transferring, the dual structure barriers of urban and rural land ownership should be broken down, and an urban-rural integrated land market should be promoted. The use value of rural construction land needs to be evaluated, and a solid foundation for the price markets of urban-rural land needs to be laid. Additionally, validating and maintaining the “qualification right” of rural residents [55] is necessary to make reasonable, legal, and compliant distribution of the homestead of farmers. It is important to adequately compensate farmers when they voluntarily surrender their idle residents. It also needs to illegalize the residents which have been sold and covered by this round of Territorial Space Planning and Rural Land System Reform.

(4) In the comprehensive land consolidation and ecological restoration of the whole region, conversion of farmlands to ponds, forests, and grasslands, and the re-development and re-utilization of inefficient-idle construction land should be implemented according to local land resource conditions. Specifically, based on the results of the Third Land Survey in China and the Third Soil Survey in China, dynamic monitoring, evaluation, and database building of the national cultivated land quality grade should be conducted at the earliest. Continuously monitoring the quantity and quality of “in-out balance” of cultivated land [3] and dynamically monitoring the quality of water bodies, such as the Nandu river, Meishe river, Dongzhaigang mangrove, and Hongcheng lake, are very important. Also, the investigation, filing, storage, market entry, and supervision of idle land in the central urban area should be performed.

## Figures and Tables

**Figure 1 ijerph-19-16902-f001:**
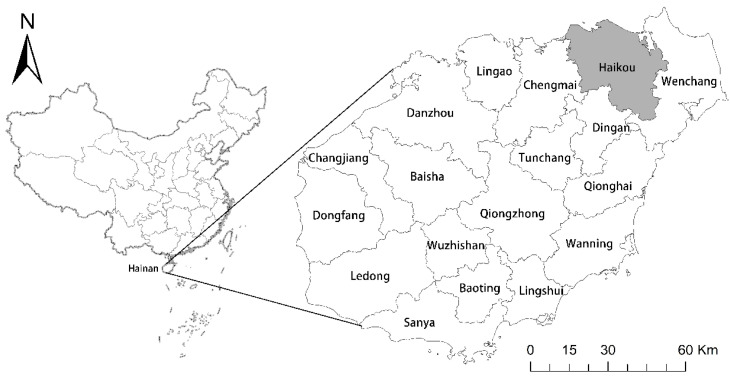
A map of the study area.

**Figure 2 ijerph-19-16902-f002:**
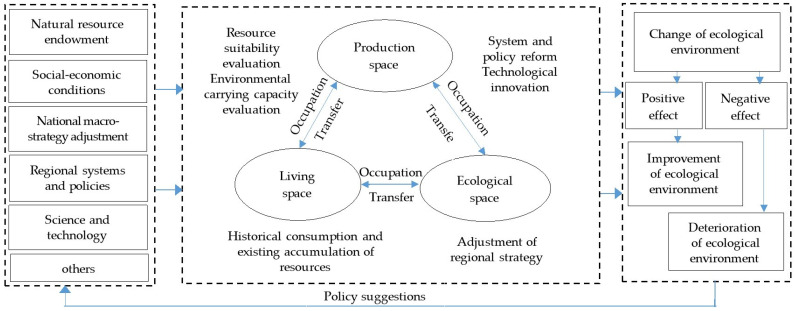
A framework of the analysis [49,50].

**Figure 3 ijerph-19-16902-f003:**
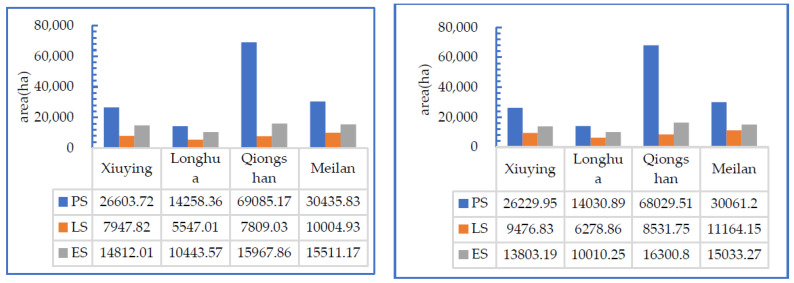
The scale of PLES in four Districts of Haikou, China, in 2009 and 2018.

**Figure 4 ijerph-19-16902-f004:**
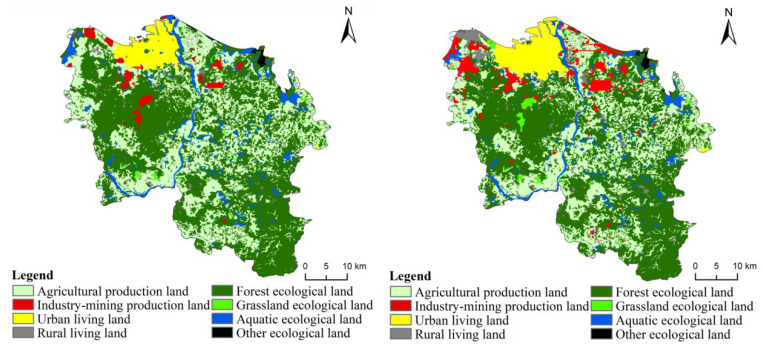
The spatial distribution pattern of land use functions in Haikou in 2009 and 2018.

**Figure 5 ijerph-19-16902-f005:**
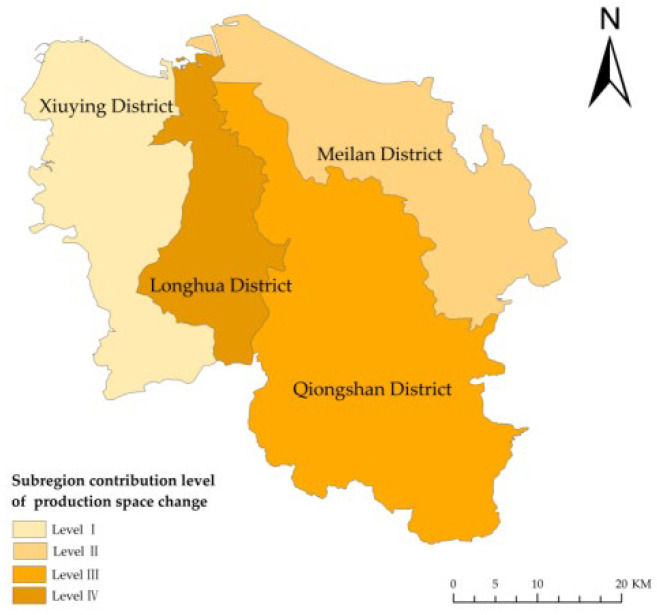
Sub-region contribution level of production space conversion and industrial-mining PS conversion in Haikou, China, from 2009 to 2018.

**Figure 6 ijerph-19-16902-f006:**
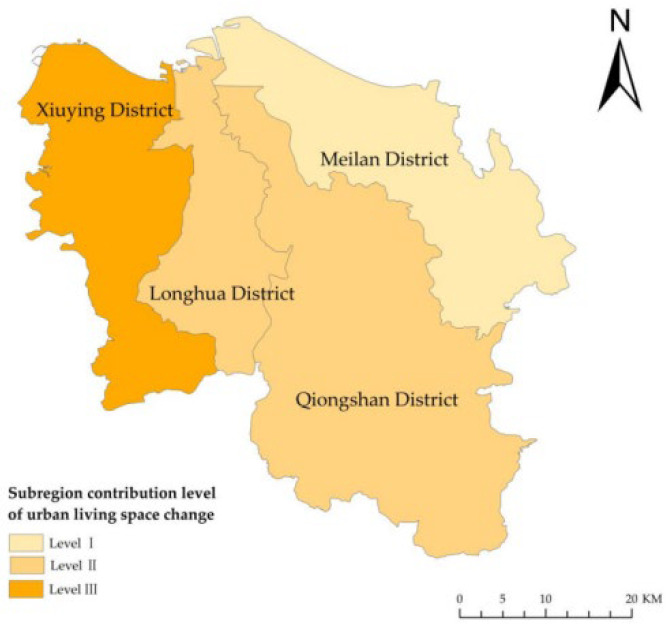
The sub-region contribution level of urban living space conversion in Haikou, China, from 2009 to 2018.

**Figure 7 ijerph-19-16902-f007:**
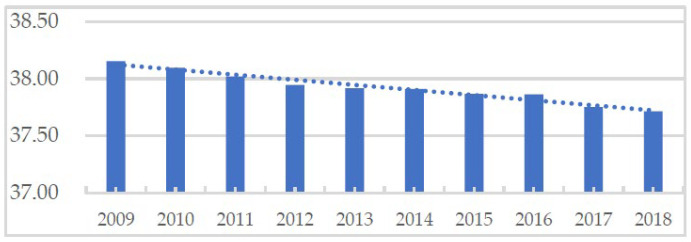
Changes in the eco-environmental quality index of Haikou from 2009 to 2018.

**Figure 8 ijerph-19-16902-f008:**
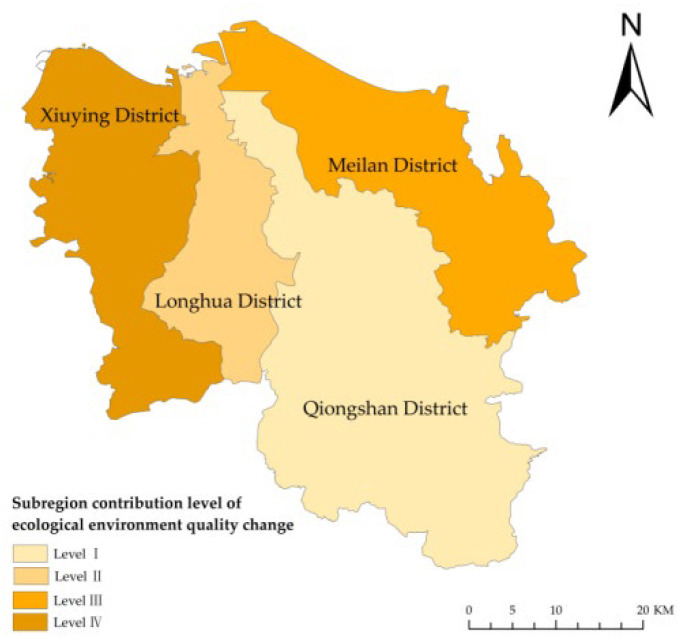
Sub-region contribution level of the change in the eco-environment quality in Haikou, China, from 2009 to 2018.

**Figure 9 ijerph-19-16902-f009:**
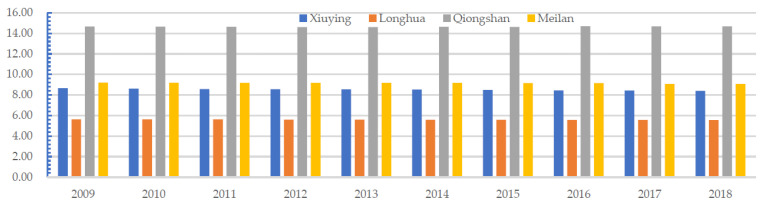
Changes in the eco-environmental quality index of the four districts in Haikou, China, from 2009 to 2018.

**Figure 10 ijerph-19-16902-f010:**
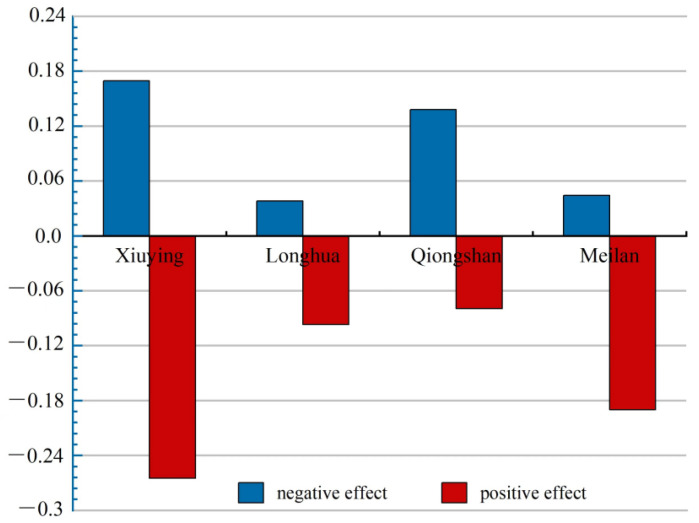
Ecological effects of land use function transition in Haikou from 2009 to 2018.

**Table 1 ijerph-19-16902-t001:** Classification of the dominant land use functions and the eco-environment quality index.

Classification of Dominant Land Use Functions		Land Use Types of Land Resources Survey	Eco-Environment Quality Index
First-Level Land Use Types	Second-Level Land Use Types		
Production land	Agricultural production land	Paddy field, dry land, garden land, reservoir water surface	0.291
	Industry-mining production land	Mining land, industry and traffic construction land, hydraulic structures	0.150
Living land	Urban living land	Urban land	0.200
	Rural living land	Rural residential land	0.200
Ecological land	Forest ecological land	Forest land, shrub land, open forest land, and other forest land	0.877
	Grassland ecological land	Natural grassland, artificial grassland, and other grassland	0.782
	Aquatic ecological land	Rivers, lakes, ditches, ponds, mudflats, and shoals	0.512
	Other ecological land	Sandy land, saline-alkali land, marshland, bare land, bare rock gravel land	0.025

**Table 2 ijerph-19-16902-t002:** Eco-environment effects of land use function transition and changes in the eco-environment quality index.

**Negative effect of eco-environment impact**	**Conversion of Land Use Types**	**Difference of Eco-Environment Quality Index**	**Positive effect of eco-environment impact**	**Conversion of Land Use Types**	**Difference of Eco-Environment Quality Index**
	Forest ecological land-Rural residential land	−0.677		Other ecological land-Grassland ecological land	0.757
	Forest ecological land-urban land	−0.677		Agricultural production land-Grassland ecological land	0.491
	Forest ecological land-Industry-mining production land	−0.727		Other ecological land-Urban land	0.175
	Forest ecological land-Agricultural production land	−0.586		Other ecological land-Rural residential land	0.175
	Agricultural production land-Industry-mining production land	−0.141		Other ecological land-Industry-mining production land	0.125
	Agricultural production land-Rural residential land	−0.091		Other ecological land-Agricultural production land	0.266
	Agricultural production land-Urban land	−0.091		Industry-mining production land-Agricultural production land	0.141
	Urban land-Industry-mining production land	−0.05			
	Aquatic ecological land-Industry-mining production land	−0.362			
	Aquatic ecological land-urban land	−0.312			

**Table 3 ijerph-19-16902-t003:** The area (in ha) of “production-living-ecological” spaces in Haikou from 2009 to 2018.

Year	Total Area	Production Space		Living Space		Ecology Space			
		Agricultural Production Land	Industry-Mining Production Land	Urban Land	Rural Residential Land	Forest Ecological Land	Grassland Ecological Land	Aquatic Ecological Land	Other Ecological Land
2009	228,426.49	130,403.97	9979.12	20,271.00	11,037.79	42,102.58	3.51	8529.94	6098.58
2010	228,426.48	129,965.90	10,319.75	20,640.95	11,095.40	41,833.56	135.83	8520.17	5914.92
2011	228,426.48	129,447.33	10,410.44	21,198.52	11,253.36	41,591.88	183.65	8513.49	5827.81
2012	228,908.73	129,295.41	10,529.99	21,884.52	11,290.93	41,487.24	179.99	8497.86	5742.79
2013	228,908.73	128,871.08	10,634.98	22,194.87	11,378.00	41,365.67	275.04	8492.35	5696.74
2014	228,908.73	128,296.93	10,820.70	22,292.20	11,627.18	41,209.91	531.04	8482.69	5648.08
2015	228,908.73	128,119.90	10,911.48	22,440.71	11,749.93	41,084.15	531.04	8470.22	5601.30
2016	228,908.73	127,229.48	11,141.28	22,935.95	11,979.66	40,861.66	910.40	8411.79	5438.51
2017	228,950.65	126,637.98	11,843.91	23,126.01	12,039.60	40,618.75	907.02	8404.49	5372.89
2018	228,950.65	126,429.72	11,921.83	23,295.38	12,156.21	40,518.89	905.31	8400.53	5322.78
2009–2018	Increment	−3974.25	1942.71	3024.38	1118.42	−1583.69	901.8	−129.41	−775.8
	Increase rate (%)	−3.05	19.47	14.92	10.13	−3.76	25,692.31	−1.52	−12.72

**Table 4 ijerph-19-16902-t004:** The land use transfer matrix of the conversion of PLES in Haikou from 2009 to 2018 (unit: ha).

2009		2018							
		Production Land		LIVING LAND		Ecological Land			
		Agricultural Production Land	Industry-Mining Production Land	Urban Living Land	Rural Living Land	Forest Ecological Land	Grassland Ecological Land	Aquatic Ecological Land	Other Ecological Land
Production land	Agricultural production land	434.24	1168.28	1491.5	662.76	0.14	792.52	0.01	5.92
	Industry-mining production land	138.97	19.94	122.57	56.49	7.13	0	0	0.75
Living land	Urban living land	5.56	96.95	25.78	0.04	2.31	0	0	0.78
	Rural living land	1.87	82.4	36.76	0	18.63	0	0	0.72
Ecological land	Forest ecological land	114.46	525.61	508.53	383.04	0.53	3.57	0	17.31
	Grassland ecological land	3.39	4.14	0.72	0.85	0	0	0	0
	Aquatic ecological land	1.92	67.96	59.95	2.67	0	0	0	0
	Other ecological land	73.05	190.16	257.56	89.55	0	115.27	0	0

**Table 5 ijerph-19-16902-t005:** Land-type conversion of LUFT and its contribution rate to the eco-environment quality of Haikou from 2009 to 2018.

**Negative effect of eco-environment impact**	**Conversion of Main Land Use Types**	**Difference of Eco-Environment Quality Index**	**Conversion Area (ha)**	**Ecological Contribution Rate**	**Proportion in Ecological Contribution Rate**
	Forest ecological land-Rural living land	−0.677	383.04	−0.11	17.74
	Forest ecological land-urban living land	−0.677	508.53	−0.15	23.55
	Forest ecological land-Industry-mining production land	−0.727	525.61	−0.17	26.14
	Forest ecological land-Agricultural production land	−0.586	114.46	−0.03	4.59
	Agricultural production land-Industry-mining production land	−0.141	1168.28	−0.07	11.27
	Agricultural production land-Rural living land	−0.091	662.76	−0.03	4.13
	Agricultural production land-Urban living land	−0.091	1491.50	−0.06	9.29
	Rural living land-Industry-mining production land	−0.05	96.95	0.00	0.33
	Aquatic ecological land-Industry-mining production land	−0.362	67.96	−0.01	1.68
	Aquatic ecological land-Rural living land	−0.312	59.95	−0.01	1.28
	Total		5079.04	−0.64	100
**Positive effect of eco-environment impact**	Other ecological land-Grassland ecological land	0.757	115.27	0.04	14.55
	Agricultural production land-Grassland ecological land	0.491	792.52	0.17	64.86
	Other ecological land-Urban living land	0.175	257.56	0.02	7.51
	Other ecological land-Rural living land	0.175	89.55	0.01	2.61
	Other ecological land-Industry-mining production land	0.125	190.16	0.01	3.96
	Other ecological land-Agricultural production land	0.266	73.05	0.01	3.24
	Industry-mining production land-Agricultural production land	0.141	138.97	0.01	3.27
	Total		1657.08	0.26	100

**Table 6 ijerph-19-16902-t006:** Transition categories and their ecological contribution rate of land use functions in the four districts of Haikou from 2009 to 2018.

Ecological Effects	Conversion of Main Land Use Types	Difference of Eco-Environment Quality Index	Conversion Area (ha)				Ecological Contribution Rate				Proportion in Ecological Contribution Rate			
			Xiuying	Longhua	Qiongshan	Meilan	Xiuying	Longhua	Qiongshan	Meilan	Xiuying	Longhua	Qiongshan	Meilan
**Negative effect of eco-environment impact**	Forest ecological land-Rural living land	−0.677	79.95	91.93	99.4	111.76	−0.0236	−0.0272	−0.0294	−0.0330	8.92	28.04	36.90	17.39
	Forest ecological land-urban living land	−0.677	311.36	67.62	56.17	73.38	−0.0921	−0.0200	−0.0166	−0.0217	34.74	20.63	20.85	11.42
	Forest ecological land-Industry-mining production land	−0.727	224.82	68.30	33.33	199.16	−0.0714	−0.0217	−0.0106	−0.0632	26.94	22.37	13.29	33.28
	Forest ecological land-Agricultural production land	−0.586	37.81	22.30	16.94	37.41	−0.0097	−0.0057	−0.0043	−0.0096	3.65	5.89	5.44	5.04
	Agricultural production land-Industry-mining production land	−0.141	518.06	101.49	8.65	432.07	−0.0319	−0.0063	−0.0005	−0.0266	12.04	6.45	0.67	14.00
	Agricultural production land-Rural living land	−0.091	116.80	119.01	143.07	283.88	−0.0046	−0.0047	−0.0057	−0.0113	1.75	4.88	7.14	5.94
	Agricultural production land-Urban living land	−0.091	609.58	275.70	297.85	308.37	−0.0242	−0.0110	−0.0118	−0.0123	9.14	11.31	14.86	6.45
	Rural living land-Industry-mining production land	−0.05	24.47	0.69	0.03	71.76	−0.0005	0.0000	0.0000	−0.0016	0.20	0.02	0.00	0.82
	Aquatic ecological land-Industry-mining production land	−0.362	41.83	0.31	3.96	21.86	−0.0066	0.0000	−0.0006	−0.0035	2.50	0.05	0.79	1.82
	Aquatic ecological land-Rural living land	−0.312	2.11	2.60	0.42	53.55	−0.0003	−0.0004	−0.0001	−0.0073	0.11	0.37	0.07	3.84
	Total		1966.79	749.95	659.82	1593.2	−0.2650	−0.0969	−0.0797	−0.1900	100	100.00	100.00	100.00
**Positive effect of eco-environment impact**	Other ecological land-Grassland ecological land	0.757	47.56	67.71	0	0	0.0208	0.0224	0.0000	0.0000	12.27	58.66	0.00	0.00
	Agricultural production land-Grassland ecological land	0.491	11.17	2.40	622.58	156.37	0.0049	0.0005	0.1335	0.0335	2.88	1.35	96.71	76.08
	Other ecological land-Urban living land	0.175	111.04	78.24	31.28	37	0.0485	0.0060	0.0024	0.0028	28.64	15.67	1.73	6.42
	Other ecological land-Rural living land	0.175	19.90	18.50	17.21	33.94	0.0087	0.0014	0.0013	0.0026	5.13	3.71	0.95	5.89
	Other ecological land-Industry-mining production land	0.125	68.92	82.06	4.32	34.86	0.0301	0.0045	0.0002	0.0019	17.77	11.74	0.17	4.32
	Other ecological land-Agricultural production land	0.266	34.48	6.24	5.16	27.17	0.0151	0.0007	0.0006	0.0032	8.89	1.90	0.43	7.16
	Industry-mining production land-Agricultural production land	0.141	94.70	43.23	0.02	1.02	0.0414	0.0027	0.0000	0.0001	24.42	6.98	0.00	0.14
	Total		387.77	298.38	680.57	290.36	0.1694	0.0382	0.1381	0.0441	100	100.00	100.00	100.00

## Data Availability

All data generated or analyzed during this study are included in this published article.

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
