# Peer review of "Process and Eco-Environment Impact of Land Use Function Transition under the Perspective of “Production-Living-Ecological” Spaces—Case of Haikou City, China"

_ijerph, 2022, doi:10.3390/ijerph192416902_

Round 1

Reviewer 1 Report

1. The term “land use change (LUC)” in line no: 44 seems to be an incomplete content mentioned, please clarify.

2. The analysis framework (presented in the section of "research methods") needs more citations to support.

3. In line no: 385, Natural resource endowment (especially soil quality), to insert a comma before “soil quality” would be better.

4. What is “qualification right” of rural residents in line no: 509, it needs citation to clarity.

5. It would be much better to make Policy implications than “Policy Suggestions” in line no: 577.

6. “in-out balance” of cultivated land in line no: 621, was not very clear, please to clarify.

7. It is necessary to clarify those terms such as land use types, land categories, land use functions categories and transition categories and the definition of land use function transition.

8. It is hard for readers to follow those abbreviations in manuscript. I suggest reduce using abbreviation.

Author Response

Thank you very much for your comments on our paper, "Process and Eco-environment Impact of Land Use Function Transition under the Perspective of “Production-Living-Ecological” Spaces-Case of Haikou City, China” (original manuscript #: ijerph-2108934). Your suggestions significantly improved our manuscript. We appriciate your insights and suggestions. Each comments has been addressed in details. 

Reviewer #1’s comments:

  1. The term “land use change (LUC)” in line no: 44 seems to be an incomplete content mentioned, please clarify.

Response: Thank you for bringing this up. Changes in land use and land cover are central to global change. In line 46, we changed it to "land use/land cover change (LUCC)."

  1. The analysis framework (presented in the section of "research methods") needs more citations to support.

Response: Thank you for bringing this up. We have added two citations to the section's The analysis framework for this point.

  1. Fu, J.; Bu, Z.; Jiang, D.; Lin, G.; Li, X. Sustainable land use diagnosis based on the perspective of production-living-ecological spaces in China, Land Use Policy, 2022, 122, 106386.
  2. Zhang, Z.; Li, J. Spatial suitability and multi-scenarios for land use: Simulation and policy insights from the production-living-ecological perspective, Land Use Policy, 2022, 119,106219.

  1. In line no: 385, Natural resource endowment (especially soil quality), to insert a comma before “soil quality” would be better.

Response: Thank you for bringing this up. In line 407, we added a comma before "soil quality."

  1. What is “qualification right” of rural residents in line no: 509, it needs citation to clarity.

Response: Thank you very much. In China, a rural resident's "qualification right" means that if you are a villager in a village, you have the right to obtain a specific area of homestead approved by the village collective for your long-term residence and use. For example, if you are a villager in a Haikou's village in Hainan Province, you can obtain a 175 m2 homestead approved by the village collective for your long-term residence and use. We have added a citation in line 636 to make it more clear.

  • Yang R, Lin Y. Rural spatial transformation and governance from the perspective of land development rights: A case study of Fenghe village in Guangzhou. Growth Change. 2022, 53(3): 1102–1121.

  1. It would be much better to make Policy implications than “Policy Suggestions” in line no: 577.

Response: Thank you very much. In line 603, we changed "policy suggestions" to "policy implications."

  1. “in-out balance” of cultivated land in line no: 621, was not very clear, please to clarify.

Response: Thank you very much. "In-out balance" of cultivated land means that cultivated land occupation and supplementation must be consistent in quantity and quality. We have added a citation to clarify this point in line 468 to help readers understand. 

  • Ma, W.; Jiang, G.; Li, W.; Zhou, T.; Zhang, R. Multifunctionality assessment of the land use system in rural residential areas: Confronting land use supply with rural sustainability demand, Environ. Manage. 2019, 231, 73–85.

  1. It is necessary to clarify those terms such as land use types, land categories, land use functions categories and transition categories and the definition of land use function transition.

Response: Thank you very much. Land use types and land categories were the same in this paper. Three first-level land use types and eight second-level land use types were included. Production land, living land, and ecological land were the three first-level land use types. Agriculture production land, industry-mining production land, urban living land, rural living land, forest ecological land, grassland ecological land, aquatic ecological land, and other types of ecological land were the eight second-level land use types. Land use dominant function categories were similarly divided into three first-level land use dominant function categories and eight second-level land use dominant function categories. We made these two terms consistent throughout the paper and changed them all to "land use types." 

In line 205, land use functions transition categories refer to the conversion of land use types (see Table 2). Land use function transition, on the other hand, denotes the conversion of the dominant land use function (reference 18).

  1. It is hard for readers to follow those abbreviations in manuscript. I suggest reduce using abbreviation.

Response: Thank you for bringing this up. We only kept the major abbreviations "LUFT" and "PLES," which are relevant to the subject of this paper. We have reduced the use of abbreviation throughout the paper.

Reviewer 2 Report

The paper explores the impact of land use function transition in the case of Haikou, China. As a first comment I suggest to insert a theoretical part (in a dedicated paragraph) with a thorough explanation of the conceptual premises: what do you mean by eco-environment impact? what differs “environment” from “eco-environment”? and more importantly what do you mean by “production-living-ecological” spaces? I find it very difficult to understand the premises for this study if not clearly positioned in a theoretical framework. 

Check the figure 2, it shows some typos and some words are still underlined in red. The formatting of table 6 is difficult to read

I am not expert in data analysis so I will leave that part out of the comments, However I would suggest to provide additional limitations in the conclusions. As a merely quantitative approach, this could leave out some impactful aspects such as social, cultural, historical ones. I understand this is not the main focus of the paper, but to fit some reflections in the conclusions could add some perspective on usability of this study and replicability in other contexts.

Author Response

Thanks a lot for your comments on our paper, "Process and Eco-nvironment Impact of Land Use Function Transition under the Perspective of “Production-Living-Ecological” Spaces——Case of Haikou City, China” (original manuscript #: ijerph-2108934). Your suggestions significantly improved our manuscript. We appreciate your insights and suggestions. Each comment has been addressed in details.

Reviewer #2’ comments:

As a first comment I suggest to insert a theoretical part (in a dedicated paragraph) with a thorough explanation of the conceptual premises: what do you mean by eco-environment impact? what differs “environment” from “eco-environment”? and more importantly what do you mean by “production-living-ecological” spaces? I find it very difficult to understand the premises for this study if not clearly positioned in a theoretical framework. 

Response: The following is a more detailed description of the analysis framework (in a dedicated paragraph) :

Population, enterprises, capital, and technologies tend to move to and accumulate in Haikou city of Hainan Province as national socio-economic development policies and strategies are adjusted, such as the establishment of a Free Trade Port and rural land system reform in Hainan Province, resulting in the conversion of different land use types. Production land occupied by living land, ecological land occupied by living land, and production land et cetera are examples. Therefore, living space oriented by living land would compress production space oriented by production land. Also, there would be a transfer from production space to living space or others. In addition, with the spatiotemporal transfer and optimization of production space, living space, and ecological space (PLES) based on resource suitability evaluation and historical consumption of land resources, a series of responses to these policies and strategies will emerge, such as transition in the land use function (LUFT), which is also determined by land use type, and changes in eco-environment quality. All of them can be attributed to various factors, such as internal conversion between agricultural and industrial-mining production land, conversion of production land to living land, and so on. Regional eco-environment quality indices were used to compare the positive and negative effects of LUFT on the environment that corresponded with socioeconomic development in specific regions over time. The LUFT is primarily determined by changes in the structure of land use types over time, including the quantity and spatial distribution of attributes. The transfer of PLES denotes changes in attributes such as regional eco-environment quality and the positive or negative effects of LUFT on the environment. (line 154175)

Two citations had been added to support the framework of analysis.

  1. Fu, J.; Bu, Z.; Jiang, D.; Lin, G.; Li, X. Sustainable land use diagnosis based on the perspective of production–living–ecological spaces in China, Land Use Policy, 2022, 122, 106386.
  2. Zhang, Z.; Li, J. Spatial suitability and multi-scenarios for land use: Simulation and policy insights from the production-living-ecological perspective, Land Use Policy, 2022, 119,106219.

Check the figure 2, it shows some typos and some words are still underlined in red.

Response: In Figure 2, the typos of “suitablity” and “Histrorical”(underlined in red) have been corrected to "suitability" and "Historical." And Figure 2 has been replaced with a new one.

The formatting of Table 6 is difficult to read.

Response: Thank you very much. We recommend that the assistant editor re-format Table 6. Table 4 would also need to be re-formatted, for example, to reduce the front size so that it is easier to read.

I am not expert in data analysis so I will leave that part out of the comments, However I would suggest to provide additional limitations in the conclusions. As a merely quantitative approach, this could leave out some impactful aspects such as social, cultural, historical ones. I understand this is not the main focus of the paper, but to fit some reflections in the conclusions could add some perspective on usability of this study and replicability in other contexts.

Response: Thank you for your thoughtful suggestion. Section 6, which is a separate section, now includes some policy implications.

.

Round 2

Reviewer 2 Report

Thank you for your work on the paper and accurate reply.

I believe the conclusions still need acknowledgement of study limitations (changing only the title of the policy part might not be proven sufficient). But I understand the intention to keep the part as it is.